# Dimensions of Self-Perceived Functionality in Older Adults Based on the Brazilian National Health Survey

**DOI:** 10.3390/ijerph22121770

**Published:** 2025-11-21

**Authors:** Jeisyane Acsa Santos do Nascimento, Thais Sousa Rodrigues Guedes, Sanderson José Costa de Assis, Clécio Gabriel de Souza, Marcelo Cardoso de Souza, Rafael Limeira Cavalcanti, Kamila Eduarda da Silva, Diego Neves de Araujo, Johnnatas Mikael Lopes, Marcello Barbosa Otoni Gonçalves Guedes

**Affiliations:** 1Graduate Program in Physical Therapy, Federal University of Rio Grande do Norte, UFRN, Lagoa Nova 59078-900, Brazil; marcelo.cardoso@ufrn.br (M.C.d.S.); rafaellimeirafisio@gmail.com (R.L.C.); marcello.guedes@ufrn.br (M.B.O.G.G.); 2Graduate Program in Collective Health, Federal University of Rio Grande do Norte, UFRN, Lagoa Nova 59078-900, Brazil; thais.guedes@ufrn.br; 3Department of Physical Therapy, Faculdade Mauricio de Nassau Natal, Av. Engenheiro Roberto Freire, 1514-Capim Macio, Natal 59080-400, Brazil; sanderson_assis@hotmail.com; 4Graduate Program in Physical Therapy, Faculdades de Ciências da Saúde do Trairi-UFRN/FACISA, Av. Rio Branco, S/N, Santa Cruz 59200-000, Brazil; clecio.gabriel@ufrn.br; 5Independent Researcher, Sapé 58340-000, Brazil; kabiila@outlook.com; 6Medical Sciences Complex, Federal University of Alagoas, Av. Manoel Severino Barbosa-Bom Sucesso, Arapiraca 57309-005, Brazil; diego.araujo@arapiraca.ufal.br; 7Department of Medicine, Federal University of Vale do São Francisco, UNIVASF, Av. José de Sá Maniçoba, S/N-Centro, Petrolina 56304-917, Brazil; johnnatas.lopes@univasf.edu.br

**Keywords:** aging, functional status, national health survey, epidemiological studies, cross-sectional studies

## Abstract

The aging population is a global phenomenon that represents a significant demographic shift, with life expectancy closely related to individuals’ functional capacity. Background/Objectives: Identify latent dimensions in self-perceived functional variables among Brazilian older adults. Methods: This cross-sectional study used data from module K of the 2019 National Health Survey (NHS) in Brazil. Multivariate statistics were applied using Principal Component Analysis (PCA). Results: The sample consisted of 22,728 older adults from all regions of Brazil, with a predominance of females (56%), an average age of 70 years (CI: 69.68–70.03), 61% engaged in weekly physical exercise, and most living with a spouse or partner (56.3%). The multivariate statistical analysis, conducted through PCA, resulted in two distinct macro-dimensions of functionality: Activities of Daily Living (ADLs) and Instrumental Activities of Daily Living (IADLs). When analyzing each macro-dimension separately, we identified internal micro-dimensions. Within ADLs, two subdimensions stood out: ADLs Upper Limb Function and ADLs Lower Limb Function. Within IADLs, the micro-dimensions included IADLs Health Management and IADLs Independent Outdoor Mobility. Close results in the subdimensions of variable K25 did not allow for a clear distinction between the estimated IADLs. However, given the importance of this variable in explaining the cognitive aspect of the functional capacity construct, we suggest maintaining it as a separate subdimension: IADLs Financial Management. Conclusions: The latent dimensions of functional capacity identified in this study may help guide functional assessment in older adults, inform therapeutic decision-making, shape public policy, and support further research on functional capacity in aging populations.

## 1. Introduction

Population aging is a global phenomenon that has been widely debated worldwide [1]. By 2030, it is estimated that one in six individuals will be 60 years or older. The proportion of people in this age group has increased from 1 billion in 2020 to 1.4 billion, and projections indicate that the global population of individuals aged 60 and over will double by 2050, reaching 2.1 billion. This demographic trend demands attention in the fields of health, economics, and social policies [2].

Although population aging initially began in high-income countries such as Japan, this demographic shift is now more pronounced in low- and middle-income countries [3]. Brazil, classified as a developing country, is experiencing exponential growth in its elderly population, which now represents 14% of the total population, categorizing it as an “aged country” [3,4].

In response to this trend, the Pan American Health Organization (PAHO) launched the Decade of Healthy Aging 2021–2030, placing older adults at the center of this intergovernmental initiative. This plan brings together governments, civil society, professionals, institutions, international agencies, and the media to monitor population aging, its findings, and challenges. The goal is to improve the quality of life for older adults, their families, and communities [5].

Increased life expectancy is an indicator of human development, reflecting socio-environmental conditions and summarizing mortality experiences within a population [6,7]. Parallel to this, there is healthy aging [8], which is understood as “the process of developing and maintaining the functional ability that enables well-being in older age” [9].

Functional capacity is influenced by intrinsic factors, environmental conditions, and their interactions within the individual [10]. Assessing functional capacity in older adults involves the performance of Activities of Daily Living (ADLs), which include self-care abilities such as bathing and eating [11], and Instrumental Activities of Daily Living (IADLs), which evaluate environmental adaptation skills such as shopping and household tasks [12]. Some assessment tools measure functional capacity as a broad construct, without considering dimensions that may be more specific to certain populations [13,14,15,16,17,18].

Stratifying the elderly population is essential for designing health interventions that address both general and individual needs. A macro-level approach enables the development of comprehensive public policies, while a micro-level approach identifies individual conditions and needs, allowing for the implementation of personalized therapeutic plans. This individualized approach is crucial for detecting early warning signs that require timely interventions, thereby preventing or minimizing functional decline. By integrating an analysis of ADL and IADL micro-dimensions, we can ensure effective interventions that promote healthy aging and improved quality of life, addressing the complexity and diversity of aging [18].

Previous studies [16,17] have demonstrated the importance of functional self-perception in the quality of life of the elderly, highlighting that this self-perception influences well-being and the ability to face daily challenges. Although functional self-perception is often assessed through self-assessment questionnaires, such as Activities of Daily Living (ADLs) and Instrumental Activities of Daily Living (IADLs) scales, these instruments are often ineffective as they do not adequately capture the complexity of the experiences of older individuals. 

Thus, this study aims to identify latent dimensions of self-perceived functionality in Brazilian older adults. This analysis is crucial for advancing research on ADL and IADL micro-dimensions, as these dimensions can reveal significant insights into the daily lives of older individuals. Understanding these micro-dimensions not only enhances knowledge about functionality in old age but also underscores the importance of incorporating disciplines such as psychology, sociology, and public policy planning in aging research. This multidisciplinary approach will enable the development of more effective and personalized public policies, fostering healthy aging and improved quality of life for the elderly population, while also guiding future research in this field.

## 2. Materials and Methods

Study Design

This is a retrospective cross-sectional study based on secondary data collected from the 2019 National Health Survey (“Pesquisa Nacional de Saúde”—NHS) in Brazil. The NHS is a population-based survey developed in collaboration with the Brazilian Ministry of Health to expand the thematic investigation of the Health Supplements of the National Household Sample Survey (PNAD) [19].

Ethical Considerations

The 2019 NHS adhered to Resolution 196/96 of the National Health Council, ensuring participants’ voluntariness, anonymity, and the right to withdraw at any time. The National Research Ethics Commission (CONEP) approval number for the 2019 NHS is 3.529.376.

Participants

This study analyzed data from participants in the 2019 National Health Survey who were 60 years or older, using Module K variables of interest, with no exclusion criteria applied.

Sampling Plan

The NHS sample was derived from a Master Sample, which is the sampling structure of the Integrated Household Survey System (SIPD). This structure consists of a set of selected units registered to facilitate subsample analyses within the SIPD, such as PNAD.

A cluster sampling design was employed in three stages: (i) Primary sampling units (PSUs) were census tracts, stratified before selection; (ii) Secondary units were households; (iii) Tertiary units were residents aged 15 years or older within the selected households.

The primary sampling unit subsample was selected using simple random sampling in both the first and second stages. Initially, IBGE (an official Brazilian Institute of Statistics) defined a fixed number of 15 households per primary group, but this approach did not ensure an equitable sample across states. Therefore, adjustments were made: 18 households were selected in Roraima, Amapá, and Tocantins; 12 households in Maranhão, Ceará, Pernambuco, Bahia, Minas Gerais, Rio de Janeiro, São Paulo, Paraná, Santa Catarina, and Rio Grande do Sul; and 15 households in all other states/federative units.

In the third stage, simple random sampling was used to select one resident aged 15 or older per household to respond to the survey questionnaire. These strategies were designed to ensure a balanced distribution of the primary household sample across Brazilian states. In addition, weighting adjustments were applied to correct for differences in selection probabilities, ensuring that the final estimates accurately reflected the national population.

Sample Size

The household and individual sample sizes were determined based on the expected precision level for estimating each indicator by domain. The number of households selected per primary sampling unit, the design effect, and the proportion of households in the target age group were analyzed to ensure representativeness.

Data Collection

The Brazilian Institute of Geography and Statistics (IBGE) organized and supervised data collection for the 2019 NHS, involving interviewers, supervisors, and coordinators from its staff. Training was initially provided to state coordinators, who then trained supervisors and interviewers. During the data collection phase, the field team received continuous online training, allowing for real-time queries and clarifications.

Survey Instrument

The results presented in this study were obtained from the 2019 NHS, conducted by IBGE, using Module K, which focuses on the health of individuals aged 60 and older. The 2019 NHS estimated that the elderly population accounted for 16.4% of the total population, totaling 34.4 million people, an increase of 8.1 million compared to the first NHS edition in 2013. The 2019 NHS comprises 26 modules, labeled alphabetically, covering various aspects of public health in Brazil. Based on the theoretical framework, only variables related to functionality from Module K were selected and analyzed, as this is the focus of this study.

Module K contains a questionnaire with 62 questions (K001–K062), assessing functional capacity, medication use, cataract occurrence, flu vaccination, and falls in individuals aged 60 and older. This study primarily evaluated functional limitations related to Activities of Daily Living (ADLs) and Instrumental Activities of Daily Living (IADLs).

The questionnaire assessed the degree of difficulty individuals experienced in performing ADLs and IADLs. The Module K variables and their corresponding questions are detailed in Table 1 below.

Data Analysis

Since this study was based on secondary data, the analysis began with data preparation. Descriptive analysis was conducted using means, standard deviations, and frequencies to characterize the sample. The latent dimensions of self-perceived functional variables were assessed through Principal Component Analysis (PCA).

The rotation method in PCA classified **D1** as ADLs, which include basic tasks such as personal hygiene, eating, and mobility. Conversely, **D2** was classified as IADLs, which involve more complex tasks requiring greater autonomy, such as financial management, public transportation use, meal preparation, and household chores.

To verify assumptions, Bartlett’s Test of Sphericity was applied. A *p*-value < 0.05 indicates sufficient correlations among variables to proceed with the analysis, allowing for the rejection of the null hypothesis [20,21]. This test was chosen considering the normal data distribution, assessed using the Kolmogorov–Smirnov Test.

Another test used to evaluate data suitability for PCA was the Kaiser–Meyer–Olkin (KMO) measure of sampling adequacy. KMO values were interpreted using a threshold of 0.80, with values closer to 1 indicating greater data suitability for PCA [22,23].

Next, the eigenvalues of the extracted factors were plotted in a Scree Plot, which identifies the inflection point of the curve [22]. This plot requires multiple factor eigenvalues to determine how many should be retained in the factor analysis, following the criterion of eigenvalues greater than 1. Components with higher eigenvalues indicate greater importance in explaining data variance, enabling a more robust interpretation of results [24].

## 3. Results

The results obtained in this study are presented in tables, figures, and charts, included in the main text and in the Appendix A. Table 2 shows the eigenvalue weights of the variables assessed in the Principal Component Analysis (PCA). Table 3 details the components, eigenvalues, variances, and accumulated variances of the PCA, as well as the Kaiser–Meyer–Olkin (KMO) sampling adequacy measure for module K. Table 4 and Table 5 display the component weights for components **1** and **2** (micro-dimensions) of the macro-dimensions ADLs (Activities of Daily Living) and IADLs (Instrumental Activities of Daily Living), respectively, through the PCA. 

### 3.1. Sociodemographic Characteristics of the Sample

The sociodemographic characteristics of the Brazilian elderly population included in the sample consisted of 22,728 individuals, 43.3% of whom were male and 56.7% female. The analysis covers several variables, including the federative unit, where a higher concentration of elderly individuals is observed in the Southeast (46.4%) and Northeast (25.4%) regions, followed by the South (15.7%), Central-West (6.4%), and North (6.1%) regions. Regarding weekly exercise duration, 61.1% of respondents reported being active. The average age of participants is 70 years (95% CI: 69.68–70.03), and the majority (81.4%) are literate. In terms of educational level, the highest level completed was the Former Primary School, with 40.4% of elderly individuals in this category. Additionally, 56.3% of the elderly stated that they have a spouse or partner living in the same household. These details are crucial for understanding the profile of the elderly population in Brazil, allowing for a more detailed analysis of their living conditions and social participation.

### 3.2. Principal Component Analysis

In Table 2, the PCA revealed two main components: **D1** and **D2**. **D1** corresponds to Activities of Daily Living (ADL), including the variables K1, K4, K7, K10, K13, K16, and K19, which have high weights, indicating a strong association with ADLs. **D2** refers to Instrumental Activities of Daily Living (IADL), including the variables K22, K25, K28, K31, and K34, which also have high weights, indicating a strong association with IADLs, which require more complexity and autonomy.

### 3.3. Component Statistics

The summary of the Principal Component Analysis (PCA) (Appendix A) presents the main components and their variances. Component **D1** is the most significant, explaining 42.3% of the total variance, with an eigenvalue of 5.07. Component **D2** explains 34.6% of the variance, with an eigenvalue of 4.15. Together, **D1** and **D2** capture 76.9% of the accumulated variance, reflecting a significant portion of the data variability.

### 3.4. Assumption Testing

The Bartlett’s Sphericity Test was also applied to identify the presence of correlations between the variables, providing statistical significance that the correlation matrix contains significant correlations between some of the variables. The significance value of this test was below 0.001 [20,21].

Another test was performed, namely the Kaiser–Meyer–Olkin (KMO) sampling adequacy measure. The overall KMO value was 0.933, indicating excellent sample adequacy for PCA. Further details can be found in Table 3.

In the scree plot, the inflection point occurred after the second component, indicating that most of the data variability was captured within the first two components. Further details can be found in Appendix A.

Table 3 presents 12 main components, of which the first two explain 76.9% of the total variance, indicating that they are the most significant and capture a large amount of variability in the data. Thus, focusing only on these two components allows for dimensionality reduction without losing relevant information, disregarding the need to analyze the remaining components.

**Table 3 ijerph-22-01770-t003:** Description of the number of components, their eigenvalues, variances, and cumulative variances according to PCA analysis and the KMO Sampling Measure of the variables from module K.

Components	Eigenvalues	% Variance	% Cumulative Variance	KMO
**1**	7.6649	63.874	63.9	0.966
**2**	1.5590	12.991	**76.9**	0.928
**3**	0.7145	5.954	82.8	0.950
**4**	0.4778	3.982	86.8	0.949
**5**	0.3451	2.876	89.7	0.960
**6**	0.3278	2.731	92.4	0.915
**7**	0.2131	1.776	94.2	0.908
**8**	0.1941	1.618	95.8	0.945
**9**	0.1616	1.346	97.1	0.952
**10**	0.1402	1.168	98.3	0.974
**11**	0.1151	0.960	99.3	0.886
**12**	0.0868	0.723	100.0	0.890
Global KMO				0.933

The bold value highlights that the first two components together capture a cumulative variance greater than 70% in the Principal Component Analysis (PCA).

### 3.5. Macrodimension—ADL

Table 4 shows which variables have the highest weights in the PCA components. Values close to zero indicate that the variable is well explained by the PCA components, revealing two micro-dimensions. The variables K7, K13, K16, and K19 group together in the first component, forming ADLs-Lower Limbs. The variables K1, K4, and K10 stand out in the second component, constituting ADLs-Upper Limbs.

**Table 4 ijerph-22-01770-t004:** Description of weights in components **1** and **2** (micro-dimensions) of the ADLs macro-dimension through PCA.

	Component		
	D1	D2	Uniqueness
K1		**0.869**	0.1812
K4	0.457	**0.803**	0.1465
K7	**0.813**	0.452	0.1350
K10	0.479	**0.764**	0.1872
K13	**0.850**	0.390	0.1258
K16	**0.882**	0.360	0.0927
K19	**0.892**	0.351	0.0806

Note: The ‘varimax’ rotation was used. The bold values represent the variables that stood out in the first (**D1**) and/or second (**D2**) component of the Principal Component Analysis (PCA).

Upon analyzing the results presented in Appendix A, it is possible to retain only the first component, which represents more than 76% of the total variance. The subsequent components can be disregarded due to the decrease in eigenvalues, indicating that the most relevant information has already been captured. This approach simplifies the analysis without compromising data quality.

### 3.6. Macrodimension—IADL

The analysis of the IADL macro-dimension (Table 5) provides new insights into the self-perception of functionality in this population. The variables K22, K31, and K34 group together in the first component, named IADLs-Independent External Mobility. In the second component, the variable K28 stands out, identified as IADLs-Health Management. On the other hand, the variable K25, IADLs-Finances, does not clearly distinguish which component exhibits the most relevant characteristics.

**Table 5 ijerph-22-01770-t005:** Description of weights in components **1** and **2** (micro-dimensions) of the IADL macro-dimension through PCA.

	Component		
	1	2	Uniqueness
K22	**0.818**	0.413	0.1606
K25	**0.638**	0.592	0.2425
K28		**0.930**	0.0535
K31	**0.896**	0.308	0.1014
K34	**0.902**		0.1049

Note: The ‘varimax’ rotation was used. The bold values represent the variables that stood out in the first and/or second component of the Principal Component Analysis (PCA).

The analysis of the eigenvalues and the variance of the principal components of the IADLs (Appendix A) reveals that the first component, with an eigenvalue of 3.802, explains 76.03% of the total variance. The second component, with an eigenvalue of 0.536, represents 10.71% of the variance. Components **3**, **4**, and **5** explain 10.71%, 6.87%, 4.08%, and 2.31% of the total variance, respectively. Thus, the first three components together account for 93.6% of the variance, indicating that most of the information from the original data is concentrated in them. These results provide new insights into the latent structure of the data, helping to reduce dimensionality without significant information loss.

## 4. Discussion

This study’s primary objective was to identify latent dimensions in self-perceived functionality variables among elderly Brazilian individuals through the 2019 National Health Survey (NHS). We identified two macro-dimensions, **D1** and **D2**, which correspond to the ADLs (Activities of Daily Living) and IADLs (Instrumental Activities of Daily Living), as outlined by the 2019 NHS questionnaire that categorizes questions about functional activities. Additionally, micro-dimensions were identified within the ADLs, such as ADLs-Upper Limbs and ADLs-Lower Limbs, as well as within the IADLs, including IADLs-Health Management, IADLs-Independent External Mobility, and IADLs-Finances. This detailed analysis of latent dimensions can foster a deeper understanding of the different self-perceived functionality profiles of elderly individuals, providing new insights for public policies aimed at disability prevention.

The two macro-dimensions, ADLs and IADLs, together explained most of the accumulated variance. These results were expected, as the 2019 NHS questionnaire lists basic and instrumental daily living activities 19. However, another finding was that within this accumulated variance, there was a distinct distribution between the ADLs and the IADLs domains. One hypothesis is that ADLs reflect simpler activities, such as eating and dressing [25], while IADLs involve more complex skills, such as shopping and going to the doctor [26].

This study identified micro-dimensions within the ADLs and IADLs, which may be relevant for the formulation of new health care strategies. Our findings may complement previous studies [27,28] focusing on public policies regarding functional capacity in older adults. The macro- and micro-dimensions of functionality presented here can guide new strategies for functional assessments that aim to promote healthy aging [29], the construction of social policies, and active participation of individuals [30].

The analyses performed on the ADLs and IADLs components revealed distinct clusters. In the ADLs dimension, we found the micro-dimensions: ADLs-Upper Limbs and ADLs-Lower Limbs. In the IADLs, we identified three micro-dimensions: IADLs-Health Management, IADLs-Independent External Mobility, and IADLs-Finances. This categorization was based on the similarities in the tasks performed by the variables that make up each micro-dimension.

The ADLs-Upper Limbs grouping, which includes variables that require greater use of the upper limbs, is supported by the Physical Performance Test (PPT) [31], a tool frequently used to assess the overall functional capacity of older adults [32]. The abbreviated version of the PPT includes seven functional tasks, such as writing a sentence, simulating eating, lifting a book and placing it on a shelf, dressing and undressing a jacket, picking up a coin from the floor, turning 360 degrees while standing, and walking 15.2 m.

These tasks encompass dimensions such as upper limb function, functional mobility, and dexterity [33], allowing for the inference of similar domains in the execution of the variables in this data group. Activities like “eating” and “dressing” are clearly outlined in the PPT, while “bathing” could be associated with other tasks, such as “lifting a book and placing it on a shelf” (related to reach), turning and/or standing, and picking objects from the floor, in addition to dexterity. This association is based on the interdependence of activities of daily living (ADLs), which demonstrates that skills acquired in one activity can influence the performance of others [12].

Another contributory factor of the PPT is its prediction of functional limitations. A study investigated an elderly population and found that the PPT predicts functional limitations in ADLs before they are self-reported by patients with chronic conditions [34]. This analysis highlights the relevance of the PPT in assessing functional capacities of the upper limbs and its importance in conjunction with other evaluative measures such as self-reports, reports, and performance tests [35], offering a more comprehensive understanding of the needs of the elderly population, considering the heterogeneity of population-based studies.

The second micro-dimension identified in the ADLs, ADLs-Lower Limbs, is crucial for understanding the functional capacities of the population of elderly individuals. Tasks performed in this context cover various domains such as mobility, self-care, and social interaction, each playing a vital role in promoting autonomy and quality of life [36,37].

However, it is essential that these tasks are not broadly evaluated under the ADLs category, as the results showed that the variables have different weights, highlighting their distinction in different components, which suggests that there are similarities that group them or specificities that separate them. A study by Millan and José [38] supports our findings by identifying that walking speed is a strong predictor of the onset of disability in ADLs. This relationship is pertinent since the skills required for tasks in this grouping demand considerable effort from the lower limbs, such as sitting and standing from a bed or chair, as well as bathroom transfers.

Despite the research on this micro-dimension, there is a gap in the literature, as studies specifically evaluating ADLs based on similarities in tasks are scarce [27,28]. This lack of investigation hampers a detailed understanding of the specific functional needs of older adults, emphasizing the need for more stratified approaches in functional assessments. The identification and analysis of micro-dimensions in ADLs can, therefore, contribute to the development of more targeted and effective interventions, promoting a more active and healthy aging process.

In this context, the Timed Up and Go (TUG) test is a functional assessment that evaluates mobility and balance [39], directly reflecting the individual’s ability to perform everyday tasks. Integrating TUG results with the identification of micro-dimensions in ADLs, especially ADLs-Lower Limbs, could provide valuable insights into the relationship between functional mobility and the performance of daily activities.

Therefore, considering micro-dimensions in ADLs not only enriches functional assessment but also supports the formulation of more specific and effective interventions, promoting active and healthy aging. This approach can significantly contribute to the development of health policies that address the individual needs of elderly individuals, rather than a one-size-fits-all perspective.

In the Instrumental Activities of Daily Living (IADLs) analysis, we identified distinct micro-dimensions: IADLs-Health Management, which stood out in the first component; IADLs-Independent External Mobility and IADLs-Finances, emphasized in the second component. These groupings were established based on the similar and/or individual characteristics of the tasks performed by each variable. This approach allows for a deeper understanding of the different dimensions that compose the IADLs, emphasizing the importance of each one in the daily lives of older adults.

Previous studies [37,40] have shown that a tendency to be disorganized and less self-disciplined correlates with limitations in both IADLs and ADLs. Moreover, personality traits, as described in the Five-Factor Model (FFM) [41], have been linked to self-reports of functionality in old age. This background justifies the grouping of IADLs-Health Management and suggests that the act of “taking medications alone” goes beyond physical and motor aspects, involving self-discipline, organization, and personality traits. This relationship underscores the importance of considering psychological and behavioral factors in assessing and intervening in the functional capacities of older adults, fostering a broader and more effective approach to promoting health in this population.

In this context, social support is a major ally in healthy aging and good self-perception of health and functionality [42]. Studies highlight the importance of the composition and extent of the social network, emphasizing that having someone close has a positive impact on the physical and mental health of elderly individuals [43,44]. This relationship suggests that social support contributes to emotional well-being but can also directly influence functional capacities, such as health management, including the ability to “take medications alone.” Therefore, considering social support is crucial when evaluating IADLs, promoting a more comprehensive and effective approach to elderly individuals’ care.

In this regard, the Guedes Tool [45] stands out as a valuable instrument for assessing informal social support among elderly individuals in Brazil. This tool covers four dimensions of social support: composition and extent of the social network, instrumental support and availability, reciprocity and longitudinality, and emotional support and social participation. This approach allows for a more complex evaluation of the social needs of older adults, contributing to the management of comprehensive care for this population and emphasizing the importance of social support in promoting healthy and functional aging.

IADLs-Independent External Mobility is vital for the autonomy of this population and is influenced by socioeconomic factors, access to health care, and education [46]. Elderly individuals with better socioeconomic conditions tend to have greater access to resources that facilitate mobility, while those in vulnerable situations face significant barriers, such as a lack of adequate transportation or instruction on how to use public transportation to access essential services. Moreover, the level of education impacts the ability to understand and use available services, as those with higher education generally have more information about transportation options and urban mobility. This relationship is corroborated by the 2019 National Health Survey (NHS) [47], which indicates that the higher the education level, the lower the limitation indicators.

Considering the interconnected factors is essential for categorizing the functional levels of older adults. However, the NHS, by applying segmented and quantitative questionnaires with predefined responses, overlooks contextual factors that affect self-perceived functionality. This limitation may lead to an incomplete understanding of the needs and challenges faced by this population, as socioeconomic conditions, accessibility, and education are significant. Therefore, it is crucial to develop assessment tools that integrate these contextual variables, offering a more holistic view of the reality of older adults.

Finally, the IADLs-Finances represents the last micro-dimension identified within the IADLs. This micro-dimension showed similar values in both the first and second IADLs components, making it less effective in distinguishing between these principal components. However, it was grouped in the second component due to its higher value, indicating relative importance within this analysis. This similarity suggests that the ability to manage finances may not be sufficiently differentiated from other dimensions, complicating the analysis of the financial functionality of older adults.

Although the mathematical expression of this IADLs-Finances variable is not particularly effective in distinguishing these main components, it remains crucial for financial management. These variables are more related to cognitive aspects than to functional practices, but are also essential for functionality, since cognitive or mental decline can lead to impairment in IADLs [48]. Therefore, additional approaches are needed to better explore and understand the cognitive importance of finances in the autonomy and quality of life of this population.

The incorporation of cognitive assessment tools, such as the Montreal Cognitive Assessment (MoCA) [49], can be fundamental for better understanding the interactions between executive functions and financial management. These assessments can help identify specific areas that require intervention, promoting greater financial autonomy and improving the quality of life for individuals. This integrated approach will enable a more comprehensive understanding of the importance of finances in daily life and mental health for this population.

In the perspective addressed here, the International Classification of Functioning, Disability, and Health (ICF) [50] is globally recognized as the framework for classifying, describing, recording, and measuring functionality and disability. Considering the 2019 NHS variables, this study identified relevant factors such as age, sex, race, and education level, which fit into the ICF components related to personal factors. Additionally, it considers contextual factors, such as social environment, household, income, and geographic region, which contribute to elderly people’s functionality. The dimensions of self-perceived functionality are divided into ADLs and IADLs. These aspects reflect the components of the ICF, which include body functions and structures, activities and participation, and contextual factors.

Despite the robustness of the applied method and the sample size studied, this study has some limitations that should be considered. The use of the Module K questionnaire from the 2019 NHS, although comprehensive, may not capture all relevant dimensions of functional capacity in older adults, particularly in areas such as cognition, psychological aspects, and social support. The inherent subjectivity of self-perception may lead to variations in results. Additionally, data collection at a single point in time, a characteristic of cross-sectional studies, prevents the assessment of changes over time and the determination of the temporal sequence of functional decline. This inability to establish causal or sequential relationships is a critical limitation, as it does not allow for understanding whether specific factors contribute to functional decline or are consequences of it. These aspects highlight the need for caution in interpreting the results and indicate that future research—addressing the comprehensiveness of the expanded concept of health and employing longitudinal designs—is essential for a better understanding of the relationships between functional self-perception and other factors relevant to the population of elderly individuals.

## 5. Conclusions

This study revealed some important latent dimensions of self-perceived functionality in elderly Brazilian individuals. We highlight the macro-dimensions of Activities of Daily Living (ADLs) and Instrumental Activities of Daily Living (IADLs), and the micro-dimensions: ADLs-Upper Limbs, ADLs-Lower Limbs, IADLs-Health Management, IADLs-Independent External Mobility, and IADLs-Finances.

Additionally, we understand that micro-dimensions emphasize the need for personalized approaches in public policies to promote active and healthy aging. This research also highlighted gaps in the evaluation of IADLs, especially in cognitive and socio-emotional aspects.

Considering functional capacity as a broad construct, this study can suggest some relevant aspects for assessing functionality in elderly individuals in future research. It is further suggested that functional evaluations of older adults should use instruments that consider not only physical performance but also contextual factors, such as socio-economic conditions, cognitive and psychological aspects, access to health care, and that incorporate the latent dimensions identified here.

Finally, the functional capacity dimensions listed here can serve as a foundation for the development of actions that prevent functional loss in elderly individuals, as well as for the creation of public policies that promote the health of these individuals in the comprehensive sense of health.

## Figures and Tables

**Table 1 ijerph-22-01770-t001:** Module K of the Brazilian National Health Survey Questionnaire, 2019—Health of Individuals Aged 60 and Older.

Question	Response
K1. In general, how much difficulty does ___ have eating alone?	1. No difficulty 2. Slight difficulty 3. Great difficulty 4. Unable to do it
K4. How much difficulty does ___ have bathing alone?	1. No difficulty 2. Slight difficulty 3. Great difficulty 4. Unable to do it
K7. How much difficulty does ___ have going to the bathroom alone?	1. No difficulty 2. Slight difficulty 3. Great difficulty 4. Unable to do it
K10. How much difficulty does ___ have dressing alone?	1. No difficulty 2. Slight difficulty 3. Great difficulty 4. Unable to do it
K13. How much difficulty does ___ have walking inside the house alone?	1. No difficulty 2. Slight difficulty 3. Great difficulty 4. Unable to do it
K16. How much difficulty does ___ have lying down or getting up from bed alone?	1. No difficulty 2. Slight difficulty 3. Great difficulty 4. Unable to do it
K19. How much difficulty does ___ have sitting down or getting up from a chair alone?	1. No difficulty 2. Slight difficulty 3. Great difficulty 4. Unable to do it
K22. How much difficulty does ___ have shopping alone?	1. No difficulty 2. Slight difficulty 3. Great difficulty 4. Unable to do it
K25. How much difficulty does ___ have managing finances alone?	1. No difficulty 2. Slight difficulty 3. Great difficulty 4. Unable to do it
K28. How much difficulty does ___ have taking medication alone?	1. No difficulty 2. Slight difficulty 3. Great difficulty 4. Unable to do it
K31. How much difficulty does ___ have going to the doctor alone?	1. No difficulty 2. Slight difficulty 3. Great difficulty 4. Unable to do it
K34. How much difficulty does ___ have going out alone using transportation?	1. No difficulty 2. Slight difficulty 3. Great difficulty 4. Unable to do it

**Table 2 ijerph-22-01770-t002:** Eigenvalue weights of the variables from Module K of the 2019 Brazilian National Health Policy, assessed by Principal Component Analysis.

	Component		
	ADL	IADL	Uniqueness
K1	**0.653**	0.345	0.455
K4	**0.728**	0.470	0.249
K7	**0.892**		0.143
K10	**0.725**	0.457	0.266
K13	**0.864**		0.169
K16	**0.891**		0.146
K19	**0.894**		0.141
K22	0.327	**0.847**	0.175
K25	0.307	**0.803**	0.261
K28	0.341	**0.676**	0.427
K31		**0.875**	0.165
K34		**0.867**	0.180

The bold values are the variables of ADLs and IADLs that stand out in the first and second components, respectively, of the Principal Component Analysis (PCA).

## Data Availability

The datasets presented in this article are not readily available, as they consist of secondary data from the Brazilian National Health Survey. Due to the sensitive nature and complexity of the data, access is restricted. Requests for access to the datasets should be directed to researcher Johnnatas Mikael Lopes at johnnatas.lopes@univasf.edu.br.

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
