# Peer review of "Dimensions of Self-Perceived Functionality in Older Adults Based on the Brazilian National Health Survey"

_ijerph, 2025, doi:10.3390/ijerph22121770_

Round 1
Reviewer 1 Report
Comments and Suggestions for Authors
Although the topic is pertinent, well-founded, using a valuable database, with a well-described methodology, I recommend a small review with settings for publication, namely:
- Explanation of hypotheses and expansion of practical recommendations, which helped in the use of the research.
- Critical reinforcement in the limitations section.

Linguistic and typing corrections:
-
- Example: "iresearchthis microdimension" (p. 10).
- "Address" appears as "Addres" in some affiliations.
Author Response
Reviewer 1: Comments and Suggestions for Authors
Although the topic is relevant, well-founded, utilizing a valuable database and a well-described methodology, I recommend a minor revision for publication settings, namely:
1. Explanation of Hypotheses and Expansion of Practical Recommendations that Assist in the Use of the Research.
I appreciate your observations and would like to emphasize that the manuscript already addresses some of the raised issues. Regarding the hypotheses, we discussed the distinction between AVDS and AIVDS, suggesting that the former reflects simpler activities, while the latter requires more complex skills, which can be considered a hypothesis to be explored. As for the practical recommendations, we mentioned that the identified macro and micro dimensions can guide new strategies for functional assessment and that social support is essential for healthy aging, suggesting interventions to strengthen support networks. Furthermore, we discussed the need to integrate contextual factors into assessments, which reflects an important practical recommendation. I am open to further suggestions to enrich the work even more.
2.Critical Reinforcement in the Limitations Section.
We appreciate your comments regarding the necessity to address the limitations of cross-sectional studies. In response, we have revised the limitations section of our study. We added a paragraph highlighting that, although the Module K questionnaire from the 2019 National Health Survey (PNS) is comprehensive, it may not capture all relevant dimensions, and the subjectivity of self-perception may lead to variations in results. Additionally, we emphasized that data collection at a single point in time prevents the assessment of changes and the determination of the temporal sequence of functional decline, limiting the ability to establish causal relationships. These points underscore the need for caution in interpreting the results and the importance of future research with longitudinal designs. This amendment can be found on page 13, paragraph 7, lines 469; page 14, paragraph 1, lines 470-473.
Comments on the Quality of the English Language
Linguistic and Typographical Corrections:
-
- Example: "iresearchthis microdimension" (p. 10).
- "Address" appears as "Addres" in some affiliations.
Thank you for pointing out the typographical error in the phrase "iresearchthis microdimension" (p. 10). I have made the necessary adjustments to correct this error. We appreciate your attention to detail and your support.
Reviewer 2 Report
Comments and Suggestions for Authors
This manuscript employs principal component analysis (PCA) on cross-sectional data from the 2019 Brazilian National Health Survey to investigate latent dimensions of self-perceived functionality among older Brazilian adults. The manuscript identifies two macro-dimensions - Activities of Daily Living (ADLs) and Instrumental Activities of Daily Living (IADLs) - along with micro-dimensions including upper limb function, lower limb function, health management, independent outdoor mobility, and financial management. These findings provide valuable insights for formulating public policies to promote healthy aging.
Here are some details identified that need to be improved:
Introduction
Page2 line84- When introducing the term "self-perceived functionality" in older adults for the first time in the introduction, it is recommended to cite relevant theoretical evidence to clarify this concept for readers (e.g., through self-assessment questionnaires) or to explain its relationship with "Activities of Daily Living (ADLs)" and "Instrumental Activities of Daily Living (IADLs)."
Materials and Methods
Page 3-In the "Sample Planning" , the adjustments to the sampling plan are not yet sufficiently detailed. Here, you could supplement the rationale behind the allocation of sample sizes across states—for example, explaining how weighting adjustments were applied to achieve a balanced distribution of samples among states.
Results
page 6-In "Table 2", the eigenvalue weights for variable K have missing entries in both the ADLs and IADLs columns, which may confuse readers. Here, it is recommended to clarify the reason for these gaps (e.g., whether they are due to low "Uniqueness" values or other methodological considerations).
Discussion
Page10line333- the article suggests a possible association between "bathing" and "lifting a book and placing it on a shelf" in the PPT, but the reasoning appears weak and lacks logical support. It is recommended to cite prior theoretical research as evidence for this claim.
Page12 line430-,the article states that "IADL – Finances" is more strongly linked to cognitive function, but the discussion remains vague without sufficient explanation of this association. It is advised to provide theoretical evidence (e.g., executive function decline affecting financial management) and suggest incorporating cognitive assessment tools in future studies to validate this dimension.
Page10 line452- while the article recommends future longitudinal research designs, it does not explicitly address the limitations of cross-sectional studies. A clear example should be provided—such as the inability to determine the temporal sequence of functional decline—to strengthen the argument.
Terms such as “elder(s),” “(the) elderly,” and “seniors” are suggested not to be used because such denominations connote discriminatory and negative stereotypes that may undercut research-based recommendations for better serving the needs of individuals and populations. More details please refer to Reframing Aging Journal Manuscript Guidelines of GSA: https://static.primary.prod.gcms.the-infra.com/static/site/gsa/document/Reframing_Aging_Journal_Manuscript_Guidelines.pdf?node=412d7ccc31fac597b9de
Author Response
|
Point-by-point response to Comments and Suggestions for Authors |
|
Comments 1: Introduction- Page2 line84- When introducing the term "self-perceived functionality" in older adults for the first time in the introduction, it is recommended to cite relevant theoretical evidence to clarify this concept for readers (e.g., through self-assessment questionnaires) or to explain its relationship with "Activities of Daily Living (ADLs)" and "Instrumental Activities of Daily Living (IADLs)."
|
|
Response 1: We appreciate your comment and valuable suggestions. We agree that it is essential to clarify the concept of "self-perceived functionality" for the readers. In response, we have made adjustments to the manuscript to better clarify this concept and its relationship with "Activities of Daily Living (ADLs)" and "Instrumental Activities of Daily Living (IADLs)." This change can be found on page 3, paragraph 7, lines 84-91.
“Previous studies (16,17) have demonstrated the importance of functional self-perception in the quality of life of the elderly, highlighting that this self-perception influences well-being and the ability to face daily challenges. Although functional self-perception is often assessed through self-assessment questionnaires, such as Activities of Daily Living (ADLs) and Instrumental Activities of Daily Living (IADLs) scales, these instruments are often ineffective as they do not adequately capture the complexity of the experiences of older individuals. Thank you for your comment and valuable suggestions. I agree that it is essential to clarify the concept of "self-perceived functionality" for the readers.”
|
|
Comments 2: Materials and Methods Page 3-In the "Sample Planning", the adjustments to the sampling plan are not yet sufficiently detailed. Here, you could supplement the rationale behind the allocation of sample sizes across states—for example, explaining how weighting adjustments were applied to achieve a balanced distribution of samples among states.
|
|
Response 2: We appreciate your comments regarding the sampling plan. The National Health Survey (PNS) employed a rigorous sampling process to ensure the representativeness of the data. Adjustments in the allocation of sample sizes were made to balance the distribution among the states, considering their demographic and geographic particularities, with weighting adjustments to accurately reflect the national population. We will add this information and the reference for the PNS in the corresponding paragraph to facilitate the reader's understanding. This change can be found on page 3, paragraph 7, lines 131-133.
“In addition, weighting adjustments were applied to correct for differences in selection probabilities, ensuring that the final estimates accurately reflected the national population”
Comments 3: Results page 6-In "Table 2", the eigenvalue weights for variable K have missing entries in both the ADLs and IADLs columns, which may confuse readers. Here, it is recommended to clarify the reason for these gaps (e.g., whether they are due to low "Uniqueness" values or other methodological considerations).
Response 3: We appreciate your observation regarding Table 2. The absence of entries for the eigenvalue weights of variable K in the ADLs and IADLs columns is a standard feature of the software used. This program omits values that are considered very low, specifically those below 0.3. This approach aims to ensure that only significant values are presented, thereby avoiding confusion and maintaining clarity in the results.
Comments 4: Discussion Page10 line333- the article suggests a possible association between "bathing" and "lifting a book and placing it on a shelf" in the PPT, but the reasoning appears weak and lacks logical support. It is recommended to cite prior theoretical research as evidence for this claim. Response 4: We appreciate your comment and recognize that the initial reasoning may have seemed weak. To address this issue, we have revised the manuscript to include references to previous theoretical research, such as that of Lawton and Brody (1969). These references support the interdependence of activities of daily living (ADLs) and demonstrate how skills acquired in one activity can influence the performance of others. This change can be found on page 11, paragraph 5, lines 345-347.
“This association is based on the interdependence of activities of daily living (ADLs), which demonstrates that skills acquired in one activity can influence the performance of others (12)”
Comments 5: Discussion Page12 line430-, the article states that "IADL – Finances" is more strongly linked to cognitive function, but the discussion remains vague without sufficient explanation of this association. It is advised to provide theoretical evidence (e.g., executive function decline affecting financial management) and suggest incorporating cognitive assessment tools in future studies to validate this dimension.
Response 5: We appreciate your comment regarding the statement that "IADL - Finances" is more strongly related to cognitive function. We acknowledge that the discussion was vague, and we have revised the manuscript to provide a clearer explanation of this association. Cognitive decline can indeed compromise the Instrumental Activities of Daily Living (IADLs), including financial management. To support this assertion, we have included theoretical evidence on how declines in executive function affect the ability to manage personal finances. Furthermore, we suggest incorporating cognitive assessment tools in future studies to validate this dimension. This change can be found on page 13, paragraph 4, lines 443-444; page 13, paragraph 5, lines 447-453.
“since cognitive or mental decline can lead to impairment in IADL (48)”; “The incorporation of cognitive assessment tools, such as the Montreal Cognitive Assessment (MoCA) (49), can be fundamental for better understanding the interac-tions between executive functions and financial management. These assessments can help identify specific areas that require intervention, promoting greater financial au-tonomy and improving the quality of life for individuals. This integrated approach will enable a more comprehensive understanding of the importance of finances in daily life and mental health for this population.”
Comments 6: Discussion Page10 line452- while the article recommends future longitudinal research designs, it does not explicitly address the limitations of cross-sectional studies. A clear example should be provided—such as the inability to determine the temporal sequence of functional decline—to strengthen the argument.
Response 6: We appreciate your comments regarding the need to address the limitations of cross-sectional studies. In response, we have revised the limitations section of our study. We added a paragraph highlighting that, while the Module K questionnaire from the 2019 National Health Survey (PNS) is comprehensive, it may not capture all relevant dimensions, and the subjectivity of self-perception may lead to variations in the results. Furthermore, we emphasize that data collection at a single point in time prevents the assessment of changes and the determination of the temporal sequence of functional decline, limiting the ability to establish causal relationships. These points underscore the need for caution in interpreting the results and the importance of future research with longitudinal designs. This change can be found on page 13, paragraph 7, lines 469; page 14, paragraph 1, lines 470-473.
“Additionally, data collection at a single point in time, a characteristic of cross-sectional studies, prevents the assessment of changes over time and the determination of the temporal sequence of functional decline. This inability to establish causal or sequential relationships is a critical limitation, as it does not allow for understanding whether specific factors contribute to functional decline or are consequences of it.”
Comments 7: Terms such as “elder(s),” “(the) elderly,” and “seniors” are suggested not to be used because such denominations connote discriminatory and negative stereotypes that may undercut research-based recommendations for better serving the needs of individuals and populations. More details please refer to Reframing Aging Journal Manuscript Guidelines of GSA: https://static.primary.prod.gcms.the-infra.com/static/site/gsa/document/Reframing_Aging_Journal_Manuscript_Guidelines.pdf?node=412d7ccc31fac597b9de
Response 7: Thank you for your comments and suggestions. In response to your feedback, we have adjusted the text to avoid using the terms "elderly," "elder," and "elders," as recommended. These adjustments are highlighted in red in the manuscript. We are committed to using language that does not convey discriminatory and negative stereotypes, ensuring that our recommendations are research-based and better address the needs of individuals and populations.
|

Round 2
Reviewer 2 Report
Comments and Suggestions for Authors
It can be considered for publication.